# On the Thermodynamic Origin of Gravitational Force by Applying Spacetime Entanglement Entropy and the Unruh Effect

**DOI:** 10.3390/e21030296

**Published:** 2019-03-19

**Authors:** Shujuan Liu, Hongwei Xiong

**Affiliations:** 1College of Science, Zhejiang University of Technology, Hangzhou 31023, China; 2Wilczek Quantum Center, Department of Physics and Astronomy, Shanghai Jiao Tong University, Shanghai 200240, China

**Keywords:** spacetime entanglement entropy, Unruh effect, gravitational force, thermodynamics, holographic principle

## Abstract

We consider the thermodynamic origin of the gravitational force of matter by applying the spacetime entanglement entropy and the Unruh effect originating from vacuum quantum fluctuations. By analyzing both the local thermal equilibrium and quasi-static processes of a system, we may get both the magnitude and direction of Newton’s gravitational force in our theoretical model. Our work shows the possibility that the elusive Unruh effect has already shown its manifestation through gravitational force.

## 1. Introduction

In the last decade, the investigation of spacetime entanglement [1,2,3,4,5,6,7,8] has given remarkable opportunities to consider the coalescence of quantum mechanics and gravitational force, although it is still unclear how to unify quantum mechanics and general relativity. Nevertheless, the concept of quantum entanglement has been found to connect closely with some fundamental properties of spacetime, such as vacuum quantum fluctuations [9,10,11], the holographic principle [12,13,14], and black holes [15,16,17,18].

The concept of quantum entanglement has already promoted our understanding of Boltzmann entropy and statistical thermodynamics [19,20,21]. For a thermodynamic system we want to study, if we consider the whole system including the external environment, the thermodynamic system is highly entangled with the external environment. In this case, the usual entropy of this thermodynamic system is in fact the entanglement entropy obtained from the reduced density matrix of this thermodynamic system [22].

In the present work, we apply both the concepts of entanglement entropy and relevant thermodynamics to consider the fundamental property of spacetime. In particular, the Unruh effect for an accelerating particle is used to consider the thermodynamic origin of gravitational force. In addition, we use a quasi-static process to consider theoretically the direction of gravitational force, which has potential application for further studies of the gravitational force for dark energy [23,24], black holes, and so on.

The paper is organized as follows. In Section 2, we give a brief introduction to the Unruh effect for the Minkowski spacetime and curved spacetime. In particular, we discuss the Unruh temperature for gravitational radiation. In Section 3, we give the finite spacetime temperature distribution of matter from the consideration of spacetime entanglement entropy and statistical thermodynamics. In Section 4, based on the consideration of the local thermal equilibrium and a quasi-static process of a system, we give an interpretation to Newton’s gravitational force and in particular the attractive characteristic. In Section 5, we consider the relativistic formula of the spacetime temperature. In the last section, we give a brief summary and discussion.

## 2. Vacuum Quantum Fluctuations and the Unruh Effect for Minkowski Spacetime and Curved Spacetime

The Minkowski spacetime can be specified by the distance between two nearby points in spacetime, given by:(1)$$ds^{2}=-c^{2}dt^{2}+dx^{2}+dy^{2}+dz^{2}.$$

Even for this flat spacetime without considering the spacetime curvature of general relativity, the Minkowski spacetime has some remarkable properties when both spacetime and quantum mechanics are considered.

The confluence of special relativity and quantum mechanics will lead to nontrivial vacuum quantum fluctuations [9,10,11]. Although we do not know the exact property of the quantum vacuum, we may assume the existence of an extremely complex and time-dependent quantum vacuum state $|\Psi_{vacuum}\rangle$ for the quantum vacuum of the Minkowski spacetime.

The usual vacuum quantum fluctuations are considered for the existence of the zero-point energy of various quantum fields. The Casimir effect [11] between two conducting metals is due to the presence of the zero-point energy of electromagnetic field. Although there are other types of zero-point energy, the conducting metals can only change the zero-point energy of the electromagnetic field in a noticeable way. Hence, the Casimir effect is about the specified vacuum quantum fluctuations due to the electromagnetic field.

Now, we turn to consider the Unruh effect in both Minkowski spacetime and curved spacetime, which originates from the vacuum quantum fluctuations and the coupling between matter and spacetime, a little similar to the Casimir effect.

### 2.1. The Unruh Effect for an Accelerating Particle in Minkowski Spacetime

The Unruh effect [17,25,26] is due to vacuum quantum fluctuations of various quantum fields. For an inertial frame of reference in the Minkowski spacetime, we consider a particle with four acceleration $a^{\alpha }=d^{2}x^{\alpha }/d\tau^{2}$ with τ the proper time. We first consider the simplest case that the particle has a specified charge so that it has a coupling with a massless scalar Bose field $\phi \left(t,r\right)$. The scalar field ϕ should satisfy the following equation in Minkowski spacetime,
(2)$$\left(-\frac{1}{c^{2}}\frac{\partial^{2}}{\partial t^{2}}+{▽}^{2}\right)\phi \left(t,r\right)=0.$$

The quantization of this scalar field leads to:(3)$$\hat{\phi }\left(t,r\right)∼\int d^{3}k\left(\hat{a}\left(k\right)f_{k}+{\hat{a}}^{†}\left(k\right)f_{k}^{*}\right).$$

Here, $f_{k}=e^{ik\cdot r-iE_{k}t/\hbar }$. $\hat{a}\left(k\right)$ and ${\hat{a}}^{†}\left(k\right)$ are the annihilation and creation operators for the mode k, respectively. The vacuum state $|0_{M}\rangle$ of the Minkowski spacetime satisfies the following property:(4)$$\hat{a}\left(k\right)|0_{M}\rangle =0$$
for all the modes denoted by k.

For the particle with four acceleration $a^{\alpha }$, we should use the Rindler coordinate [27,28,29,30] to consider the expansion of the field operator $\hat{\phi }\left(t,r\right)$. In this case, we have:(5)$$\hat{\phi }\left(t,r\right)∼\int d^{3}k\left(\hat{b}\left(k\right)g_{k}+{\hat{b}}^{†}\left(k\right)g_{k}^{*}\right).$$

It is worthwhile to point out that in this case, $\hat{a}\left(k\right)\neq \hat{b}\left(k\right)$ and $f_{k}\neq g_{k}$ for nonzero $a^{\alpha }$.

For this accelerating particle, it will seem that there are excitations of the ϕ field in the Minkowski spacetime because $\langle 0_{M}|{\hat{b}}^{†}\left(k\right)\hat{b}\left(k\right)|0_{M}\rangle \neq 0$. It is shown by Unruh [17] that:(6)$$\langle 0_{M}|{\hat{b}}^{†}\left(k\right)\hat{b}\left(k\right)|0_{M}\rangle ∼\frac{1}{e^{E_{k}/k_{B}T_{U}}-1},$$
with:(7)$$T_{U}=\frac{\hbar a}{2\pi ck_{B}}$$
the so-called Unruh temperature. Here, the proper acceleration *a* in this equation is the magnitude of the four acceleration defined by:(8)$$a=\sqrt{\eta_{\mu \nu }a^{\mu }a^{\nu }}.$$

$\eta_{\mu \nu }$ is the metric of the Minkowski spacetime. Further works have verified that there are no hidden correlations in the excitations of the ϕ field, which means that the excitations are purely thermal [28].

We should note that $T_{U}$ can be only observed by this accelerating particle. Hence, only at the location of this particle, there are observable thermal excitations of the ϕ field, because only at the location of this particle, there is coupling with the ϕ field in the vacuum. It is similar to calculate $\langle 0_{p}|{\hat{a}}^{†}\left(k\right)\hat{a}\left(k\right)|0_{p}\rangle$ with $|0_{p}\rangle$ defined by $\hat{b}\left(k\right)|0_{p}\rangle =0$ for all modes k. For an observer at rest in the Minkowski spacetime, this means that this observer will think that there is a temperature distribution around the accelerating particle with peak temperature given by $T_{U}$.

### 2.2. The Unruh Effect for Curved Spacetime

The concept of Unruh temperature had been generalized to curved spacetime. This is shown clearly in [30] by Jacobson where the Unruh temperature in curved spacetime is used to give a simple derivation of Hawking temperature. Here, we give a brief introduction of the Unruh temperature for curved spacetime.

We first consider a particle with a specified charge so that it has a coupling with the ϕ field. For a curved spacetime given by:(9)$$ds^{2}=g_{\mu \nu }dx^{\mu }dx^{\nu },$$
we may also define the four acceleration in curved spacetime for this particle. The four velocity $u^{\alpha }$ is:(10)$$u^{\alpha }=\frac{dx^{\alpha }}{d\tau }.$$

The four acceleration $a^{\alpha }$ is then:(11)$$a^{\alpha }=u^{\mu }D_{\mu }u^{\alpha }.$$

Here, $D_{\mu }$ is a covariant derivative operator.

The magnitude of the four acceleration is:(12)$$a=\sqrt{g_{\mu \nu }a^{\mu }a^{\nu }}.$$

It is worthwhile to point out that the proper acceleration *a* is an invariant quantity for any observer.

In a local inertial frame of reference, it is clear that previous analysis of the Unruh effect and Unruh temperature is valid, and hence, Equation (Equation 7) can be applied to curved spacetime by replacing *a* given by Equation (Equation 8) with Equation (Equation 12). It is by the application of Equations (Equation 7) and (Equation 12) that the Hawking temperature can be derived with the Unruh effect.

### 2.3. The Unruh Effect for Gravitational Field $h_{\mu \nu }$

For a small deviation from the Minkowski metric, we may write:(13)$$g_{\mu \nu }=\eta_{\mu \nu }+h_{\mu \nu }.$$

To leading order, we get the Ricci curvature tensor as follows,
(14)$$R_{\mu \nu }=-\frac{1}{2}\left(\partial^{2}h_{\mu \nu }-\partial_{\mu }\partial_{\lambda }h_{\nu }^{\lambda }-\partial_{\nu }\partial_{\lambda }h_{\mu }^{\lambda }+\partial_{\mu }\partial_{\nu }h_{\lambda }^{\lambda }\right)+O\left(h^{2}\right).$$

Here, $h_{\mu }^{\lambda }=\eta_{\mu \nu }h^{\nu \lambda }$. Obviously, there is gauge freedom in $h_{\mu \nu }$. Similar to the case of electromagnetic field, we may use the gauge condition to consider further the physical significance of $h_{\mu \nu }$. Using the following harmonic gauge condition:(15)$$\partial_{\mu }h_{\nu }^{\mu }=\frac{1}{2}\partial_{\nu }h,$$
the symmetric tensor $h_{\mu \nu }$ will have six free components.

With this harmonic gauge condition, in a vacuum, the Einstein field equation simplifies to:(16)$$\left(-\frac{1}{c^{2}}\frac{\partial^{2}}{\partial t^{2}}+{▽}^{2}\right)h_{\mu \nu }=0.$$

After the harmonic gauge condition, we may still make a “residual” gauge transformation so that the solution becomes:(17)$$h_{\mu \nu }=\epsilon_{\mu \nu }\sin\left(\eta_{\alpha \beta }k^{\alpha }x^{\beta }+\varphi \right),$$
with:(18)$$\epsilon_{\mu \nu }=\left(\begin{array}{cccc} 0 & 0 & 0 & 0\\ 0 & \epsilon_{+} & \epsilon_{\times } & 0\\ 0 & \epsilon_{\times } & -\epsilon_{+} & 0\\ 0 & 0 & 0 & 0 \end{array}\right).$$

Here, $\epsilon_{+}$ and $\epsilon_{\times }$ represent two independent degrees of polarizations of gravitational waves. Similar to the quantization of electromagnetic waves, we obtain gravitons, after we quantize gravitational waves. Of course, the above discussions are about the weak field approximation, which can be applied in the present work.

By quantizing the $h_{\mu \nu }$ field and carrying out almost identical calculations, we will get the same Unruh temperature for the gravitational radiation. Of course, these considerations can be applied to the Unruh temperature for electromagnetic field as well. In the following sections, we will use Equation (Equation 7) as the effective temperature for gravitational radiation. It is well known that energy is the “charge” of the gravitational field $h_{\mu \nu }$. Hence, for any particle with a≠0, there is always nonzero Unruh temperature for gravitational radiation.

We want to emphasize two properties of the Unruh effect as follows.

$T_{U}$ should be regarded as a peak value of a local temperature distribution in an inertial frame of reference.Besides the case of an electrically-charged particle usually considered for the Unruh effect, the particle may have other types of charges. Hence, $T_{U}$ may also mean the temperature for other gauge fields, such as the gravitational field. Because the gravitational field is universal for any particle, Equation (Equation 7) can be applied to the gravitational field. In this paper, the Unruh temperature is considered mainly for the gravitational field.

The purpose of this subsection is to show that for an accelerating observer, it will think that there is excitation of gravitons with the same Unruh temperature as that of the scalar field and electromagnetic field. We will show that this result gives us the chance to have close connection between the Unruh effect, spacetime temperature, and gravitational force.

## 3. Finite Spacetime Temperature Distribution Due to Matter

### 3.1. The Spacetime Quantum Fluctuations

When the sum of all zero-point energies is considered, it is well known that the vacuum energy density $\epsilon_{V}$ is extremely high and even divergent as a result of a rough consideration. Usually, the finite value of $\epsilon_{V}$ may be assumed by setting the Planck energy and Planck length as the cutoff of the quantum spacetime [27]. It is natural that this will lead to violent quantum fluctuations of the spacetime geometry [31,32] at the microscopic scale of $l_{p}$. To distinguish the vacuum quantum fluctuations introduced in the preceding section, we call it spacetime quantum fluctuation in this paper.

To give a clear picture of spacetime quantum fluctuations, we consider $f_{AB}$ defined by:(19)$$f_{AB}=\sqrt{\frac{\langle \Psi_{vacuum}|d_{AB}^{2}|\Psi_{vacuum}\rangle -{\left|\langle \Psi_{vacuum}|d_{AB}|\Psi_{vacuum}\rangle \right|}^{2}}{\langle \Psi_{vacuum}|d_{AB}^{2}|\Psi_{vacuum}\rangle }}.$$

Here, $d_{AB}$ is an operator in an inertial frame of reference to measure the spatial distance between nearby points *A* and *B* in spacetime. It is clear that $f_{AB}$ shows the fluctuations of spacetime geometry.

If *A* and *B* are macroscopically separated, it is expected that the fluctuation $f_{AB}$ is negligible, while below or of the order of a microscopic distance $l_{p}$, there would be significant fluctuations in $f_{AB}$. At the present stage, we do not know the exact value of $l_{p}$. However, the existence of spacetime quantum fluctuations [31,32] and the stable spacetime property at macroscopic scales means that there should be a distance $l_{p}$. We will give further discussion of $l_{p}$ in the next section.

### 3.2. Spacetime Entanglement Entropy and Spacetime Temperature

We consider a sphere of radius *R*. The surface of this sphere divides the whole universe into two systems $S_{A}$ and $S_{B}$, i.e., the interior of the sphere $S_{A}$ and the external environment $S_{B}$. Without the presence of any other matter in the Minkowski spacetime, the entanglement entropy is [33]:(20)$$S_{entangle}=-\mathrm{Tr}\left[\rho_{A}\log\rho_{A}\right].$$

Here, $\rho_{A}={\mathrm{Tr}}_{B}\left(\rho \right)$ is the reduced density matrix with $\rho =|\Psi_{vacuum}\rangle \langle \Psi_{vacuum}|$ the density matrix for the pure state $|\Psi_{vacuum}\rangle$ of the Minkowski spacetime. It is easy to show that $S_{entangle}=-\mathrm{Tr}\left[\rho_{B}\log\rho_{B}\right]$ with $\rho_{B}={\mathrm{Tr}}_{A}\left(\rho \right)$.

For the situation that $R>>l_{p}$, it is expected that the entanglement entropy $S_{entangle}$ depends only on the property of $|\Psi_{vacuum}\rangle$ in the region of a thin spherical shell with the width of the order of $l_{p}$. Hence, it seems reasonable to assume the following conjecture of the spacetime entanglement entropy:(21)$$S_{entangle}∼\frac{A_{area}}{l_{p}^{2}}.$$

Here, $A_{area}=4\pi R^{2}$ is the area of the sphere. This is the so-called area laws for the entanglement entropy [34,35,36]. We have another way to understand this relation. From Equation (Equation 21), we may also regard $A_{area}/l_{p}^{2}$ as the number of Planck areas on the spherical surface. We will show that $l_{p}$ is the Planck length in due course. It is worthwhile to point out that at the present stage, this formula does not mean directly the holographic principle because we do not consider the possible presence of matter distribution inside the sphere yet.

Now, we consider the case that there is a classical particle with mass *M* inside the sphere. Of course, the coupling between this particle and spacetime will lead to a change of $|\Psi_{vacuum}\rangle$ on the spherical surface. Hence, the modified entanglement entropy for the sphere becomes:(22)$$S_{entangle}^{M}∼\frac{A_{area}}{l_{p}^{2}}+\Delta S_{M}.$$

For the usual case that the particle *M* only gives a slight change to the curvature of the spacetime, it is expected that $\Delta S_{M}<<S_{entangle}$. However, the presence of this particle will lead to an important effect by applying the holographic principle. The holographic principle [12,13,14] implies that the energy $Mc^{2}$ will show its effect on the spherical surface. Combined with the first law of thermodynamics $dU=T_{M}dS$, we have:(23)$$Mc^{2}∼k_{B}T_{M}\left(R\right)\times \frac{A_{area}}{l_{p}^{2}}.$$

$T_{M}\left(R\right)$ is the effective spacetime temperature on the spherical surface. From the above equation, we have:(24)$$T_{M}\left(R\right)∼\frac{c^{2}l_{p}^{2}}{4\pi k_{B}}\frac{M}{R^{2}}.$$

There is another way to understand this formula. We consider an ideal case that the mass *M* is distributed uniformly on a surface of a sphere with a radius a little smaller than *R*. Assume that the spacetime temperature of this case is the same as the case we are considering. On the spherical surface, the energy within the spatial cell of area $l_{p}^{2}$ is:(25)$$\epsilon =\frac{Mc^{2}}{4\pi R^{2}/l_{p}^{2}}.$$

Assume the microscopic freedom of this cell is *i*; we have:(26)$$\frac{ik_{B}T_{M}}{2}=\epsilon .$$

Because it is expected that *i* is of the order of one, we will also get $T_{M}$ given by Equation (Equation 24). From the result given by Equation (Equation 24), we see that our consideration is self-consistent by assuming the ideal distribution of *M* on the spherical surface. In a sense, this distribution of $T_{M}$ is the well-known Gaussian law. Here, we give an interpretation of the Gaussian law from the holographic principle and thermodynamics. In Section 5, we will give another method to calculate $T_{M}$.

It is worthwhile to discuss the following properties of this effective spacetime temperature.

(1) In the usual case, this effective spacetime temperature is extremely small by noticing that there is a factor $l_{p}^{2}$ in the above equation.

(2) This effective spacetime temperature is about the spacetime and gravitational field, rather than the electromagnetic field.

(3) Because this effective spacetime temperature originates from the entanglement entropy and the presence of *M* inside the sphere, its finite value does not mean that there would be various radiations spontaneously. We may notice these radiations only when we have an appropriate means to experience the entanglement entropy. This is a little similar to the observation of the Casimir effect [11]. We must have two conducting metals to show the Casimir effect through the coupling with the fluctuating electromagnetic field in the quantum vacuum.

It seems that it would be extremely challenging to observe this effective spacetime temperature. However, combined with the physical picture of the Unruh effect, we will show the possibility that the simultaneous considerations of this effective spacetime temperature and the Unruh effect just lead to gravitational force.

## 4. Newtonian Gravitational Force Derived by the Consideration of Local Spacetime Thermal Equilibrium

### 4.1. Spacetime Thermal Equilibrium

We consider another fictitious particle with mass *m* and assume that this particle does not have any other interaction in addition to gravitational force. The particle *M* establishes an effective vacuum temperature field $T_{M}\left(r\right)$ given by Equation (Equation 24). Now, we consider the case that the particle *m* is fixed at location r, relative to *M*. Because there is no relative motion between *M* and *m*, the whole system has the chance to be in spacetime thermal equilibrium. For simplicity, we consider the case that M>>m. To be in spacetime thermal equilibrium, there should be another effective temperature $T_{m}$ for *m* so that:(27)$$T_{M}\left(r\right)=T_{m}.$$

When the relative location between *M* and *m* is fixed, we know that in the local inertial frame of reference for *m*, *m* has a finite acceleration. It is clear that the Unruh temperature should be calculated in a local inertial frame of reference. Hence, omitting the high-order term for the proper acceleration *a*, the Unruh temperature for *m* is:(28)$$T_{m}\left(a\right)=\frac{\hbar \left|a\right|}{2\pi ck_{B}}.$$

Here, $a=d^{2}r/dt^{2}$. We will give the exact value of $T_{m}$ in Section 5. It is clear that both $T_{M}$ and $T_{m}$ are about gravitons, so that this equation is universal for any particle. This is one of the motivations of the analysis of the Unruh effect for gravitons in Section 2.3.

The spacetime thermal equilibrium condition (Equation 27) leads to:(29)$$\left|a\right|=\alpha \frac{c^{3}l_{p}^{2}}{2\hbar }\frac{M}{r^{2}}.$$

The coefficient α can be absorbed in the definition of Newton’s gravitational constant *G*. Compared with Newton’s law of gravitational force, we have:(30)$$l_{p}={\left(\frac{2\hbar G}{\alpha c^{3}}\right)}^{1/2}.$$

We see that with the choice of α=2, we get the conventional gravitational constant *G* if we regard $l_{p}$ as the Planck length. Here, we show the possibility that $l_{p}$ is more fundamental than *G* in a sense.

In the units with ℏ=1 and c=1, we have $G=l_{p}^{2}$. We see that *G* decreases with the decreasing of $l_{p}$. This is due to the fact that with the decreasing of $l_{p}$, the degree of freedom increases, and hence, the effective spacetime temperature decreases on the spherical surface. The spacetime thermal equilibrium condition means that *m* has smaller acceleration, and equivalently smaller *G*.

### 4.2. Quasi-Static Process to Determine the Direction of Gravitational Force

Previous studies only give the magnitude of gravitational force. Now, we turn to consider the direction of gravitational force. We consider a quasi-static process by an external force $F_{ext}$ so that the system is always in quasi-thermal equilibrium. In addition, we consider the case that the particle *m* moves toward *M* in a quasi-static way. Because $T_{M}∼1/r^{2}$, we see that the particle *m* will exchange heat energy with spacetime during the quasi-static process, while the kinetic energy will not change. The first law of thermodynamics then gives:(31)$$dU_{m}=\delta Q+\delta W=0.$$

Here, δQ is the heat energy absorbed from spacetime, while δW is the work by the external force on the particle *m*. δW is given by:(32)$$\delta W=F_{ext}\cdot dr.$$

Hence, for the quasi-static process of the system with $dU_{m}=0$, we have:(33)$$F_{ext}\cdot dr=-\delta Q.$$

For simplicity, we consider the case that the particle *m* moves along the line connecting *M* and *m*. If their distance increases, δQ<0, and we have:$$F_{ext}∼\frac{r}{r^{3}}.$$

This determines the direction of the external force to maintain the thermal equilibrium or time-independent location of the particle *m*. We see that this is equivalent to the fact that the gravitational force is attractive.

If we consider the case that the particle *m* moves toward *M* along the line connecting *M* and *m*, we have δQ>0. We will still have $F_{ext}∼r/r^{3}$, and this leads to the attractive characteristic of gravitational force as well. Combined with Equation (Equation 29), the gravitational force can then be written as:(34)$$F_{g}=-\frac{GMm}{r^{3}}r.$$

### 4.3. Free-Fall Motion

In a gravitational field, we know that the free-fall motion has no acceleration at all, based on Einstein’s general relativity. In this case, the Unruh temperature is zero for the particle *m*, while the spacetime temperature due to *M* is larger than zero. Hence, during the free-fall motion, there is always a temperature difference between $T_{M}\left(r\right)$ and the Unruh temperature $T_{m}$. Because of this temperature difference, the free fall motion is not a quasi-static process. This temperature difference leads to the possibility of energy exchange between spacetime and the particle *m*.

Similarly to the analysis of Joule expansion in thermodynamics, for the free-fall motion from *A* to *B*, we may construct a quasi-static process from *A* to *B* by a fictitious external force, and then, at the end of this quasi-static process, the work of the external force is given to the particle *m*. In this case, in the non-relativistic approximation, the work by the gravitational force on the particle *m* during the free-fall motion is:(35)$$\Delta W=\phi \left(r_{1}\right)-\phi \left(r_{2}\right),$$
with ϕ(r)=−GMm/r and ΔW the work done on the particle *m* by the gravitational field.

## 5. Relativistic Formula of the Spacetime Temperature $T_{M}$ of a Classical Particle

In Section 3.2, based on spacetime entanglement entropy, the holographic principle, and thermodynamics, we get the non-relativistic approximation of the spacetime temperature $T_{M}$ for a classical particle with mass *M*. It seems that it is extremely difficult to give a method to calculate $T_{M}\left(r\right)$ in the frame of general relativity because we do not know the exact mechanism to unify general relativity and quantum mechanics yet. However, in this section, we will provide the method to calculate $T_{M}\left(r\right)$ by using the local thermal equilibrium condition $T_{M}\left(r\right)=T_{m}$.

For a classical particle with mass *M*, the Schwarzschild metric is:(36)$$ds^{2}=-\left(1-\frac{2GM}{c^{2}r}\right)c^{2}dt^{2}+{\left(1-\frac{2GM}{c^{2}r}\right)}^{-1}dr^{2}+r^{2}\left(d\theta^{2}+\sin^{2}\theta d\phi^{2}\right).$$

The four-dimensional coordinate of another particle *m* is:(37)$$x^{\alpha }=\left(t,r,\theta ,\phi \right).$$

We consider the situation that the particle *m* has a fixed location, i.e., *r*, θ, and ϕ are time independent. The four velocity $u^{\alpha }$ is:(38)$$u^{\alpha }=\frac{dx^{\alpha }}{d\tau }=\left(\frac{1}{c}{\left(1-\frac{2GM}{r}\right)}^{-1/2},0,0,0\right).$$

The four acceleration $a^{\alpha }$ is:(39)$$a^{\alpha }=u^{\mu }D_{\mu }u^{\alpha }=\left(0,\frac{MG}{r^{2}},0,0\right).$$

From this result, we have:(40)$$a^{2}=g_{\mu \nu }a^{\mu }a^{\nu }={\left(\frac{MG}{r^{2}}\right)}^{2}{\left(1-\frac{2GM}{c^{2}r}\right)}^{-1}.$$

The proper acceleration is then:(41)$$a=\frac{MG}{r^{2}}{\left(1-\frac{2GM}{c^{2}r}\right)}^{-1/2}.$$

For this particle *m* with fixed *r*, θ, and ϕ, relative to *M*, the Unruh temperature is then:(42)$$T_{m}\left(a^{\alpha }\right)=\frac{\hbar G}{2\pi ck_{B}}\frac{M}{r^{2}}{\left(1-\frac{2GM}{c^{2}r}\right)}^{-1/2}.$$

We see that the factor ${\left(1-\frac{2GM}{c^{2}r}\right)}^{-1/2}$ is the correction due to general relativity.

From the local thermal equilibrium condition given by Equation (Equation 27), the relativistic formula for the spacetime temperature due to particle *M* is then:(43)$$T_{M}\left(r,\theta ,\phi \right)=\frac{l_{p}^{2}}{2\pi k_{B}}\frac{Mc^{2}}{r^{2}}{\left(1-\frac{2GM}{c^{2}r}\right)}^{-1/2}.$$

Here, for a comparison with Equation (Equation 24), we have used the Plank length $l_{p}$ in this equation. Of course, if we regard the particle *M* as a point particle, this formula only holds for the situation of $r>2GM/c^{2}$. Compared with Equation (Equation 24), we see that the factor ${\left(1-\frac{2GM}{c^{2}r}\right)}^{-1/2}$ is the correction due to curved spacetime.

Compared with the calculations of the spacetime temperature $T_{M}$ in Section 3.2, in this section, we give significant improvement to calculate $T_{M}$ by using Equations (Equation 27) and (Equation 42). These improvements also show that the calculations in Section 3.2 are valid in the semi-relativistic approximation, and hence, this gives the support for the concept of the entanglement entropy for spacetime and the relevant thermodynamics for spacetime based on this entanglement entropy. It also implies the validity of the holographic principle and relevant thermodynamics based on the concept of entanglement entropy, because the Unruh effect does not depend on the holographic principle.

## 6. Potential Application to Modified Gravity

Although the present work is not a modification to Einstein’s general relativity, we may consider the potential application to modified gravity in the future. Here, we consider the potential application to two modified theories of gravity as follows.

In [37,38,39,40], the modification of the inertia originated from a reconsideration of the quantum effect in the Unruh effect was considered to give a modified gravitational law, which has potential application to modified Newtonian dynamics [41]. In particular, in [39,40], the modified inertia due to the consideration of the long-wavelength of the order of the Hubble scale in Unruh radiation is used to explain the Pioneer anomaly [42]. This means that when the long-wavelength excitation of gravitons is considered, there may be significant modification to spacetime temperature considered in the present work, because the result of the present work relies on the local thermal equilibrium condition. When the long-wave mode is addressed, there is even the condensation of gravitons, similar to Bose–Einstein condensed gases.

Another potential application would be the case of massive gravity [43]. The observations of gravitational waves have given strong confinement on the graviton mass that it should be no more than $7.7\times 10^{-23}$ eV/c^2^ [44], which means that the Compton wavelength of the graviton would be at least $1.6\times 10^{16}$ m. This suggests that our theory will not give a significant modification to massive gravity for the long-wave mode below $1.6\times 10^{16}$ m. However, when the cosmology evolution is addressed, we cannot exclude the possibility of significant modification due to massive gravity. Another possibility is the modification of the massive gravity to black holes [45,46,47,48,49,50], which has seen intensive studies in the last few years. Near the horizon of a black hole, the Unruh effect has close connection with Hawking radiation, and it would be interesting to consider the gravitational radiation in the Unruh effect and the relevant spacetime temperature in this work.

## 7. Summary and Discussion

In summary, we consider the thermodynamic origin of Newton’s gravitational force by considering simultaneously the spacetime entanglement entropy, the holographic principle, thermodynamics, and the Unruh effect for an accelerating particle. Different from previous works on the thermodynamic origin of gravitational force [2,51,52,53], in this work, we emphasize the quantum entanglement of spacetime. In the present paper, we do not use the assumption of the displacement entropy ΔS∼d in [2,52], which implies a more solid basis for the thermodynamic origin of Newton’s gravitational force.

Currently, the direct observation of the Unruh effect is still elusive. For an electronically-charged particle, the Unruh temperature is too small to have observable electromagnetic radiation with current techniques. Most recently, a pioneering quantum simulation of coherent Unruh radiation [54] was observed based on an ultra-cold atomic system. Of course, this observation does not show directly the original Unruh effect for spacetime. In the present work, however, we show the possibility that the original Unruh effect has already shown its manifestation through gravitational force.

The purpose of this paper is to try to propose the possibility that there exists a spacetime temperature due to the curvature of spacetime because of the existence of matter. We give the general method to calculate the spacetime temperature of a classical particle by applying the thermodynamics of spacetime and the Unruh effect. Our theory suggests that the magnitude of the four acceleration for a fixed location is of equivalent importance, compared with the scalar curvature. By analyzing both the local thermal equilibrium and quasi-static processes of a system, we may give the microscopic interpretation of the attractive characteristic of classical particles, while in general relativity, this is imposed by observation [55]. It is worthwhile to point out that even in the pioneering work about the thermodynamic origin of gravitational force in [2], there is no consideration on the direction of gravitational force. In future work, we may consider the application of the spacetime temperature to black holes, e.g., the excitation of atoms falling into a black hole [56] because of the presence of the spacetime temperature in this work. Another future work may consider the influence of the spacetime temperature on the quantum correlation of matter, which would be a complementary of the recent scenario where quantum correlations are considered theoretically to affect the gravitational field [57], by emphasizing the quantum thermodynamic characteristic of work [58].

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
