# Peer review of "On the Thermodynamic Origin of Gravitational Force by Applying Spacetime Entanglement Entropy and the Unruh Effect"

_entropy, 2019, doi:10.3390/e21030296_

Round 1

Reviewer 1 Report

In this paper, the authors derived Newton's law by using thermodynamic relations, e.g., the entanglement entropy and the Unruh effect.

However, all assumptions are very loosely related. For example, the entanglement entropy is proportional to the area, but the proportional constant depends on the cutoff. Why should it be a certain 'universal' constant?

Also, they considered kT x A ~ Mc^2, where M is the mass surrounded by a sphere. However, why do we need to assume that kT x A is related to the mass which is surrounded by a sphere? This implicitly assumed the Gauss-law which already assumed 1/r^2 type potential.

They assumed the Unruh temperature, but how can we define the 'temperature'? Usually, the meaning comes from the expectation value of the number operator. In order to define the expectation value of the number operator, we need to assume the spacetime structure, which already assumes the spacetime structure, e.g., general relativity.

So, I think that this paper is too heuristic and based on circular logic. I do not recommend this paper for a publication of this journal.

Author Response

Response to Reviewer 1 Comments

Point 1:  The entanglement entropy is proportional to the area, but the proportional constant depends on the cutoff. Why should it be a certain 'universal' constant?

Response 1: We acknowledge that this proportional constant is a widely believed conjecture based on the cutoff by the Planck length and Planck energy. We add a sentence in the first paragraph of section 3.1 and reference [26]. See also the discussion below Eq. (26). In our work, the purpose of the entanglement entropy is used to calculate the effective spacetime temperature. To make this result become more reliable, we add two different methods. See Eqs. (21), (22) and relevant interpretation, and also see the added new section 5. Of course, from the derivation of Newton’s gravitational law, it is shown that this “universal” constant is relevant to gravitational constant. At least, for small deviation from the Minkowski spacetime, it is expected that there exists this “universal” constant. It is possible that in some extremal cases, this would not be a certain “universal” constant any more, and may lead to the possibility of the modification to Einstein’s field equation. However, this is not the purpose of the present work.

Point 2:Also, they considered kT x A ~ Mc^2, where M is the mass surrounded by a sphere. However, why do we need to assume that kT x A is related to the mass which is surrounded by a sphere? This implicitly assumed the Gauss-law which already assumed 1/r^2 type potential.

Response 2:The reason is due to the so-called holographic principle. To make it become more reliable, we add two new methods to calculate the spacetime temperature. See Eqs. (21), (22) and relevant interpretation, and also see the added new section 5.

Point 3:They assumed the Unruh temperature, but how can we define the 'temperature'? Usually, the meaning comes from the expectation value of the number operator. In order to define the expectation value of the number operator, we need to assume the spacetime structure, which already assumes the spacetime structure, e.g., general relativity.

Response 3:We have revised significantly the introduction of the Unruh temperature to answer this question. See section 2.

Point 4:I think that this paper is too heuristic and based on circular logic.

Response 4:Of course, this paper is not a modification to general relativity. However, from the quantum thermodynamic consideration of spacetime, we give something more about gravitational force. We give an interpretation of the attractive characteristic of gravitational force, while in general relativity, it is imposed by observation. Hence, although our work is a little heuristic, but not based on circular logic. We added the last paragraph in the last section. In a sense, our work shows that the spacetime temperature is of equivalence importance, compared with the scalar curvature. 

Reviewer 2 Report

The paper is set out to deduce the “quantum thermodynamics” origin of gravitational force by applying entanglement entropy and Unruh effect. However, it is quite superficial and I find it difficult to see where quantum thermodynamics enters into the picture. Where those fluctuation comes in. What is the length scale that motivated the search of quantum effect? The authors mentioned that previous studies only give the magnitude of gravitational force without citing them.

In view of the material presented, I cannot recommend article for publication in Entropy. 

Author Response

Response to Reviewer 2 Comments

Point 1:  However, it is quite superficial and I find it difficult to see where quantum thermodynamics enters into the picture.

Response 1: We give a brief introduction of the concept of quantum thermodynamics to show the reason why we think that we use quantum thermodynamics in the present work. See “The relevant studies have led to the rise of the field of quantum thermodynamics. In a sense, quantum thermodynamics is the study of the emergence of thermodynamic behavior for a subsystem of a composite quantum system, and hence emphasis the relation between two independent physical theories: thermodynamics and quantum mechanics.” in section 1. Ref. [22] is added so that one may consider this issue further. 

Point 2:Where those fluctuation comes in. What is the length scale that motivated the search of quantum effect?

Response 2:Those fluctuations come from the vacuum quantum fluctuations. The length scale to search the quantum gravitational effect is usually believed to be the Planck length. However, the present work shows that it determines the direction of gravitational force. We have rewritten section 2 to make the picture of those fluctuations become more clear.

Point 3:The authors mentioned that previous studies only give the magnitude of gravitational force without citing them.

Response 3:We added the last paragraph in the last section to discuss this and give the relevant references. 

Reviewer 3 Report

The authors try to develop the quantum thermodynamic origin of the gravitational force of matter by applying the spacetime entanglement entropy and the Unruh effect originating from vacuum

quantum fluctuations. However, the problem see from the paper is the naive style for the development of the calculations. What I mean is that the authors have really to develop the calculations by using the standard techniques of Quantum Field Theory. Another detail is that the Unruh temperature is defined with an invariant scalar acceleration, and initially is not clear if the authors have this concept right. What I can see from the paper is just the combination of known expressions for deriving possible new results but they are not well justified nor derived in a rigorous way. Although the paper might have some interesting points to be considered, I think that the authors must work really hard for getting the publication. I recommend to derive the results in a more rigorous way by using the techniques of QFT and if the authors use the Unruh effect, please explain a little more and develop calculations and not only write other known results. It is not clear from the paper what is fundamental and what is derivation. What is assumption, etc.

If the authors work more on the paper in agreement with these points, then I am willing to review the paper again.   

Author Response

Response to Reviewer 3 Comments

Point 1:  However, the problem see from the paper is the naive style for the development of the calculations. What I mean is that the authors have really to develop the calculations by using the standard techniques of Quantum Field Theory. Another detail is that the Unruh temperature is defined with an invariant scalar acceleration, and initially is not clear if the authors have this concept right. What I can see from the paper is just the combination of known expressions for deriving possible new results but they are not well justified nor derived in a rigorous way. Although the paper might have some interesting points to be considered, I think that the authors must work really hard for getting the publication. I recommend to derive the results in a more rigorous way by using the techniques of QFT and if the authors use the Unruh effect, please explain a little more and develop calculations and not only write other known results. It is not clear from the paper what is fundamental and what is derivation. What is assumption, etc.

Response 1: All these questions are about the physical picture and relevant QFT calculations of Unruh temperature. We have rewritten section 2 to answer all these questions by detailed considerations to Unruh temperature for Minkowski spacetime, curved spacetime and gravitational field, respectively. We also added section 5 to deepen the relevant calculations.

Round 2

Reviewer 2 Report

The authors have attempted to improve the manuscript from the first version. However, the crucial question regarding "quantum thermodynamics" has not been dealt with. In my opinion, the section 4.2 does not have any quantum description but the basic first law of thermodynamics. The authors should have in mind that "work is not an observable" in quantum. I suggest that either the authors do the calculation using the standard quantum field theory or by presenting the analogy gravity with thermodynamics to explain the Unruh effect in clear form.

Author Response

Response to Reviewer 2 Comments (Round 2)

Point 1:  I suggest that either the authors do the calculation using the standard quantum field theory or by presenting the analogy gravity with thermodynamics to explain the Unruh effect in clear form.

Response 1:We acknowledge that the application of quantum thermodynamics in this paper is a subtle issue. To avoid potential confusion, suggested by the referee, we delete the presentation of the application of quantum thermodynamics in this paper, to emphasis the thermodynamic origin of gravitational by the application of the Unruh effect. The title is also revised. We also give more detailed introduction to the Unruh effect for gravitational effect, so that the relevant standard field theory is clearer. See Sec. 2.3.

Reviewer 3 Report

The authors have added some content in this new version. What I can see is that the authors are assuming a temperature in connection to the gravitational field (waves). The authors assume that such temperature corresponds to the Unruh one. This is already a strong assumption. Besides this, the gravitational waves analysis is incomplete and there is no clear connection done by the authors connecting the waves with the Unruh temperature. Then let"s say that the authors just did this assumption. In addition, the authors never specify which gauge are they analyzing when they write the Gravitational waves equation. If the authors assume that the Unruh temperature corresponds to the one of the gravitational field, then it is evident that the gravitational force will appear hidden inside the acceleration invariant. The authors also assume perfect quantization of the gravitational field. The authors in addition assume the result for the entanglement entropy. In conclusion, I think that the authors make too many assumptions for the community to believe these results. Then the paper needs more reviews and derivations before being considered for publication. The theory of Quantized Inertia would be ideal for the authors to take a look at, where the Unruh effect is used as a fundamental principle for the derivation of gravity. It is also known as  McCulloch theory of gravity and it does not necessarily satisfies the equivalence principle. Please take a look at Mon.Not.Roy.Astron.Soc.376:338-342,2007 (arXiv:astro-ph/0612599). Take also a look to other references like arXiv:1004.3303 [astro-ph.CO].  The authors should compare their findings with those obtained in these suggested references as well as other related references.  

Author Response

Response to Reviewer 3 Comments (Round 2)

Point 1:  Besides this, the gravitational waves analysis is incomplete and there is no clear connection done by the authors connecting the waves with the Unruh temperature. Then let"s say that the authors just did this assumption. In addition, the authors never specify which gauge are they analyzing when they write the Gravitational waves equation.

Response 1: We give a much more detailed introduction to gravitational wave, gauge, quantization and the connection to Unruh effect. See Sec. 2.3.

Point 2:  The theory of Quantized Inertia would be ideal for the authors to take a look at, where the Unruh effect is used as a fundamental principle for the derivation of gravity. It is also known as  McCulloch theory of gravity and it does not necessarily satisfies the equivalence principle. Please take a look at Mon.Not.Roy.Astron.Soc.376:338-342,2007 (arXiv:astro-ph/0612599). Take also a look to other references like arXiv:1004.3303 [astro-ph.CO] The authors should compare their findings with those obtained in these suggested references as well as other related references. 

Response 2: In the last section, we add a discussion about these references and a brief comparison with our work.

Round 3

Reviewer 2 Report

The authors have added some more discussion and clearly removed the claim of deducing the Newton’s gravitational force from quantum thermodynamics. However, it is not quite clear to this referee the purpose of section 2.3 in the revised manuscript if no connection is made to it. Why is the influence of gravitational waves not included in the Unruh’s temperature? Or won’t it be more formal to derive the Unruh’s temperature from this perspective.

Again, is Eq. 19 new or taken from somewhere else? I further suggest that the second paragraph in the introduction should be rewritten as no quantum thermodynamics is done in the manuscript. 

I believe that attending to the points above will improve the manuscript.

Author Response

Response to Reviewer 2 Comments

Point 1:  However, it is not quite clear to this referee the purpose of section 2.3 in the revised manuscript if no connection is made to it. Why is the influence of gravitational waves not included in the Unruh’s temperature? Or won’t it be more formal to derive the Unruh’s temperature from this perspective.

Response 1:Section 2.3 is extended in the revised manuscript under the request of another referee to show more clearly the application of Unruh effect to gravitational waves. In this section, after more detailed description to the solution of gravitational waves, the comparison with the scalar field in Sec. 2.1 is clearer, so that it is natural to understand that the Unruh temperature can be also used for gravitational waves. This result is one of the basis of the present work, because we want to show the close connection between Unruh effect and gravitational force. See the added last paragraph in Sec. 2.3 and two sentences below Eq. (28).

Point 2: Again, is Eq. 19 new or taken from somewhere else? 

Response 2: Eq. 19 is used to show the correlation of spacetime in this paper. I am not complete sure whether this equation is used in other references.

Point 3: I further suggest that the second paragraph in the introduction should be rewritten as no quantum thermodynamics is done in the manuscript. 

Response 3: I have revised the second and the third paragraphs in this version based on this suggestion.

Reviewer 3 Report

The authors have improved the content of the paper and they have considered the changes suggested before. An additional suggestion for the authors would be to explain how to derive the gravitational effects by starting from the Unruh effect, but this time for the case where we want to obtain Modified Gravity based on Massive gravity. For a guidance about possible changes in the predictions o the Black-Hole evaporation, you can follow the following references:

1). Universe 4 (2018) no.2, 27.

2). EPL 109 (2015) no.1, 10002.

3). Eur.Phys.J. C77 (2017) no.8, 501.

4). Phys.Lett. B772 (2017) 553-558.

5). arXiv:1808.07829 [gr-qc]

6). JHEP 1104:042,2011.

7)Phys.Rev. D86 (2012) 024030.

8). arXiv:1810.07388

In these previous papers you will find relevant aspects connected to the Black-Hole evaporation process in massive gravity, including the methods of path-integral as well as Bogoliubov coefficients. Since the Unruh effect is connected to the Black-Hole evaporation, then it becomes to be important to mention this part in the paper. 

In summary, let's assume that we want to derive the gravitational interaction by using the Unruh effect, but this time  based on a modification of gravity where the graviton becomes massive. What changes should we expect in the calculations for this purpose? 

Please explain this part in a new section. I can suggest to start by understanding how the invariant-acceleration is modified in such cases. 

This part will attract more readers and it will clarify other aspects and the paper for sure will be more interesting. In addition it will show how can we make possible generalizations of the results illustrated by the authors. I think that the topic is interesting.

Author Response

Response to Reviewer 3 Comments

Point 1:  In summary, let's assume that we want to derive the gravitational interaction by using the Unruh effect, but this time  based on a modification of gravity where the graviton becomes massive. What changes should we expect in the calculations for this purpose? 

Please explain this part in a new section. I can suggest to start by understanding how the invariant-acceleration is modified in such cases. 

Response 1: I have added a new section based on this suggestion. See Section 6. The discussion on the modification to the inertia is also moved to this new section two.